# A Case of Purpura Annularis Telangiectodes of Majocchi after Anti-SARS-CoV-2 Pfizer-BioNTech Vaccine: Is There an Association?

**DOI:** 10.3390/vaccines10111972

**Published:** 2022-11-21

**Authors:** Francesca Ambrogio, Carmelo Laface, Giorgia Sbarra, Raffaele Filotico, Girolamo Ranieri, Chiara Barlusconi, Aurora De Marco, Gerardo Cazzato, Domenico Bonamonte, Paolo Romita, Caterina Foti

**Affiliations:** 1Section of Dermatology, Department of Biomedical Science and Human Oncology (DIMO), University of Bari “Aldo Moro”, 70124 Bari, Italy; 2Interventional and Medical Oncology Unit, IRCCS Istituto Tumori “Giovanni Paolo II”, 70124 Bari, Italy; 3Section of Pathology, Department of Emergency and Organ Transplantation (DETO), University of Bari “Aldo Moro”, 70124 Bari, Italy

**Keywords:** mRNA vaccine, anti-SARS-CoV-2 vaccine, pigmented purpuric dermatoses, Majocchi’s disease

## Abstract

The advent of vaccines has drastically reduced the incidence, morbidity, and mortality related to COVID-19, and with the increase in the number of vaccinated subjects, there have been reports of some adverse events, including skin reactions. In this paper, we report a clinical case of Purpura Annularis Telangiectodes of Majocchi following a third-dose administration of the Pfizer-BioNTech COVID-19 vaccine. Almost 30 days after the third dose, the patient presented erythematous annular patches on the lower limbs with purpuric peripheral areas and a central clearing with no other symptoms. A dermoscopic examination showed capillaritis, reddish-brown dot-clods on a coppery-red background caused by leaky capillaries. To date, the causes of Majocchi’s disease are not well-defined; in the literature, three vaccination-related cases have been reported: one after a flu vaccination and two after an anti-SARS-CoV-2 one. Dermatologists should be trained to promptly recognize these clinical manifestations after vaccination, which will likely become a common finding in daily clinical practice, especially given the large diffusion of SARS-CoV-2 vaccinations.

## 1. Introduction

The advent of vaccines has drastically reduced the incidence, morbidity, and mortality related to COVID-19 [1]. Globally, as of 7:26 pm CEST, 10 October 2022, there have been 618,521,620 confirmed cases of COVID-19, including 6,534,725 deaths, reported to the WHO. As of 4 October 2022, a total of 12,723,216,322 vaccine doses have been administered [2]. Obviously, with the increase in the number of vaccinated subjects, there have been reports of some adverse events, including skin reactions [3]. Among the latter, more or less frequent reactions have been reported [4], mostly edema, erythema, tenderness at the site of injection, rash, urticaria, angioedema, herpes zoster, morbilliform eruption, pityriasis rosea, vesicular rash, chilblains-like, purpuric rash, and cutaneous vasculitis, but also rare and/or anecdotal reactions such as systemic lupus erythematosus, the flaring of guttate psoriasis, erythema multiforme, lichen planus, and reactions to dermal fillers [5].

On the other hand, there is Purpura annularis telangiectodes (PAT) of Majocchi, a less common variant of Pigmented purpuric dermatoses (PPDs), a group of benign, chronic, and relapsing disorders [6,7]. To date, several clinical variants have been described, but all of them share similar histopathologic characteristics [8].

With regard to the history of PAT of Majocchi, it was first described in 1896 [8] and usually affects adolescents and young adults, especially women. The most important clinical sign corresponds to the appearance on the skin of bluish to red annular macules in which dark-red telangiectatic puncta appear; subsequently, they have a centrifugal extension with a central progressive resolution and slight atrophy, giving them an annular configuration. The eruption begins bilaterally on the lower limbs; then, it extends to the upper extremities, but this is also described as a rare unilateral form [9,10]. These lesions may be scarce or very numerous; patients can be asymptomatic or complain of a mild itch or burning. Regarding etiopathogenesis, there is no known risk factor for PAT, but some authors have suggested the possible role of viral infections [10].

Treatments are available to improve clinical manifestations, but the responses are variable [11]. However, since most of the clinical descriptions are based on isolated cases or small series, there is no sufficient evidence to support the use of specific treatments as first-line therapy. To date, the causes of Majocchi’s disease have not been well-defined; in the literature, three vaccination-related cases have been reported: one after flu vaccination [12] and two after administering an anti-SARS-CoV-2 vaccine [12,13].

In this paper, we report a clinical case involving a histopathological correlation with an unusual purpuric reaction following a third-dose administration of the Pfizer-BioNTech COVID-19 vaccine.

## 2. Case Report

In February 2022, we were notified of a 46-year-old man with a purpuric spread on both legs. He had no personal history of any disease and/or risk factor for PAT but had a positive anamnesis of a vaccination with the third dose of the Pfizer-BioNTech COVID-19 vaccine one month before. He received the first two doses of the vaccine without suffering any adverse events (AEs). Physical examination revealed erythematous annular patches on his lower limbs with purpuric peripheral areas and central clearing (Figure 1). There were no other symptoms or mucocutaneous involvement. A dermoscopic examination (Figure 2) showed capillaritis, reddish-brown dot-clods on a coppery-red background caused by leaky capillaries. Due to inflammation, blood cells may pass through small gaps between endothelial cells, resulting in petechial hemorrhage.

Complete blood cell count, basic metabolic panel (kidney function, liver function, and blood sugar), complement (C3 0.91 g/L, C4 0.21 g/L), C-reactive protein (5 mg/L), D-dimer levels (70 ng/mL), sedimentation rate (3 mm/h), coagulation tests (aPTT 30 s, PT 1.03 Ratio, and fibrinogen 209 mg/dL), and IgE were within normal ranges. Antinuclear anti-body, rheumatoid factor, HIV, Parvovirus B19-DNA PCR, Parvovirus B19 IgG-IgM, and hepatitis studies were negative.

After obtaining informed consent, a biopsy was performed, and the histopathologic examination showed chronic inflammatory infiltrates, mostly lymphomononuclear, in perivascular areas, with mild spongiosis and extravasation of erythrocytes, which confirmed the diagnosis of a Purpuric Pigmented Dermatosis (PPD), namely, PAT of Majocchi disease (Figure 3). Indeed, at a greater magnification, the lymphomonocitary infiltrate was mainly distributed around the blood vessels of the superficial dermis with the occasional presence of siderophages (Figure 4). Furthermore, it was possible to appreciate a moderate extravasation of red blood cells among collagen fibers (Figure 5).

Furthermore, we conducted an immunohistochemical investigation to determine the presence of CD4 and CD8 T-cells (with a monoclonal antibody against CD4 and CD8, Dako Agilent, 1:500 dilution) that showed a similar distribution of the lymphocytes with a perivascular and periadnexal localization (Figure 6).

The application of topical steroids (propionate clobetasol) twice a day for two weeks and then daily for two further weeks induced the resolution of the cutaneous manifestations in our patient.

## 3. Discussion

In this paper, we described the clinical signs of an adult male patient that developed PAT of Majocchi almost 30 days after a third dose of the Pfizer-BioNTech COVID-19 vaccine. The temporal proximity between the vaccine administration and the cutaneous signs suggests a possible correlation between the two events. This is also supported by the lack of other clinical features and a history of concurrent diseases.

To the best of our knowledge, only three clinical cases of pigmented purpuric dermatoses such as Majocchi’s disease have been reported as a chronological consequence of vaccination: one after a flu vaccination [14] and two after an anti-SARS-CoV-2 one [12,13]. Therefore, although etiopathogenesis is not yet well-understood, cell-mediated immunity has been implicated as a probable factor in all three cases [4,15]. The possible explanation is that the immunological stimulus due to the vaccine might be a trigger for the development of capillaritis. In detail, the Pfizer-BioNTech COVID-19 vaccine consists of viral mRNA that leads to the production of SARS-CoV-2 spike proteins after administration. It is possible that this dermatosis is caused by an immune dysregulation following the vaccination, similar to leukocytoclastic vasculitis flares [16]. For instance, according to the studies reviewed in a systematic review [17], the new development of the previously unseen lesions can be traced back to either a vaccine-related delayed hypersensitivity reaction or a T-cell-mediated reaction arising from viral molecular similarity to the cells of the skin. This immune cross-reactivity and hypersensitivity to vaccine components could lead to endothelial damage and erythrocyte extravasation, thus causing PPD.

The remote possibility of developing this cutaneous disease must not obstruct vaccine administration, including among those patients who developed PPD after a dose. Indeed, one patient who presented this cutaneous manifestation following the first dose of vaccination did not present any disease flare after the second dose of the same vaccine [12].

## 4. Conclusions

In conclusion, some papers have described the development of a new onset of pigmented purpuric dermatoses such as PAT of Majocchi following vaccination, as well as with the Pfizer-BioNTech COVID-19 vaccine. We do not want to describe Majocchi’s dis-ease as a negative adverse effect of SARS-CoV-2 vaccines; we only want to emphasize the possibility that the two events are related. We cannot prove in any way a cause–effect correlation between these two events but, as suggested by other similar cases already published in literature, the temporal correlation, the rapid clinical resolution, and the un-known exact pathogenesis of Majocchi disease may increase the suspicion for a connection between vaccination and this dermatological condition. Dermatologists should be trained to promptly recognize these clinical manifestations after vaccination, which should likely become a common finding in daily clinical practice, especially given the large diffusion of SARS-CoV-2 vaccination.

## Figures and Tables

**Figure 1 vaccines-10-01972-f001:**
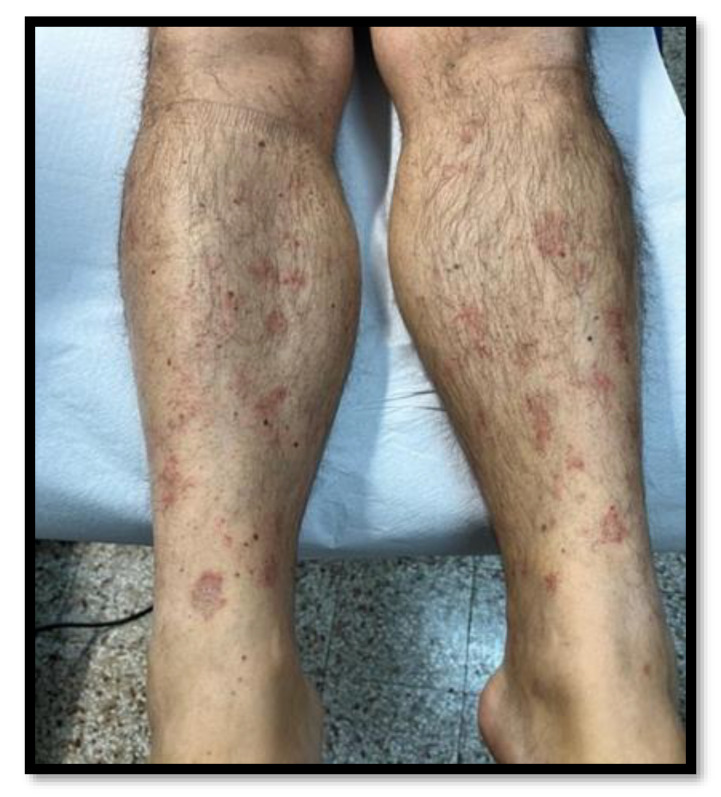
Erythematous annular patches on lower limbs with purpuric peripheral areas and central clearing.

**Figure 2 vaccines-10-01972-f002:**
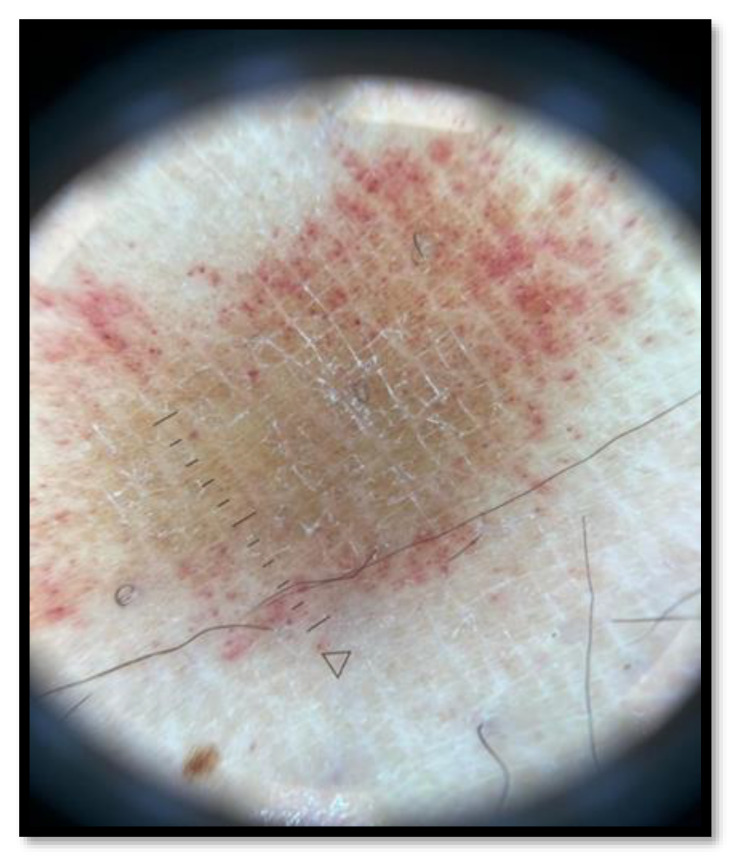
Dermoscopy: Capillaritis, reddish-brown dots-clods on a coppery-red background caused by leaky capillaries. Due to inflammation, blood cells may pass through small gaps between vessal walls’ cells, resulting in petechial hemorrhage.

**Figure 3 vaccines-10-01972-f003:**
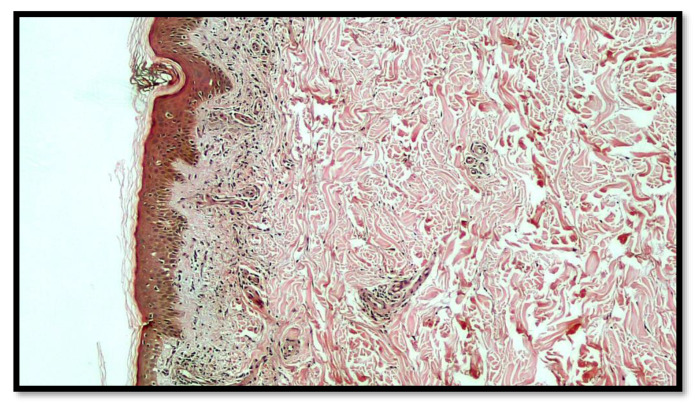
Histological preparation showing a moderate inflammatory infiltration, mainly constituted by lymphocytes and monocytes, with a perivascular distribution, and with rare and focal involvement of the epidermis that presented mild spongiosis (Hematoxylin-Eosin; Original Magnification 4×).

**Figure 4 vaccines-10-01972-f004:**
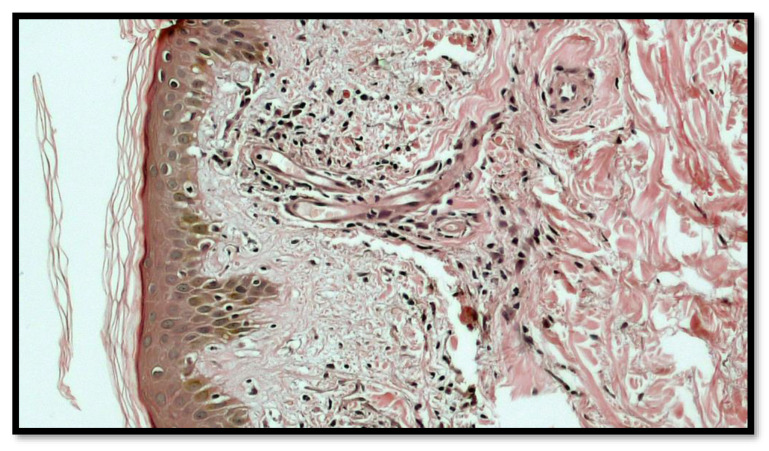
Histological preparation showing a greater magnification of the lymphomonocitary infiltrate, which was mainly distributed around the blood vessels of the superficial dermis with occasional presence of siderophages. (Hematoxylin-Eosin; Original Magnification 10×).

**Figure 5 vaccines-10-01972-f005:**
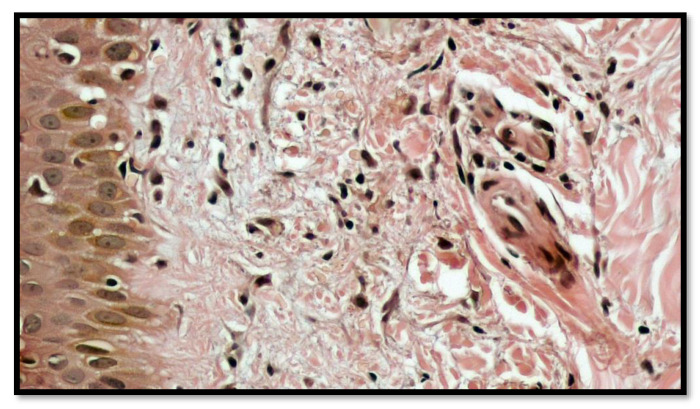
Detail at greater magnification that shows the presence of erythrocytes between the collagen fibers of the superficial dermis (Hematoxylin-Eosin; Original Magnification 20×).

**Figure 6 vaccines-10-01972-f006:**
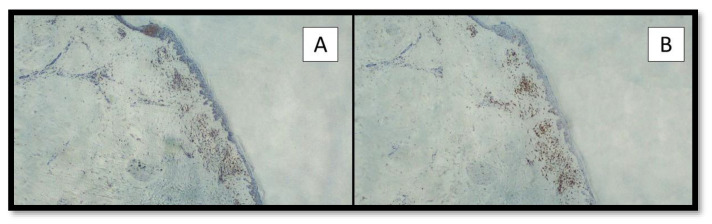
Immunostaining for CD4 (**A**) and CD8 (**B**) T-lymphocytes: note the similar quantity of T-cells and distribution at perivascular and peri-adnexal sites. (Immunohistochemistry; Original Magnification 4×).

## Data Availability

Not applicable.

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
