# Peer review of "A Case of Purpura Annularis Telangiectodes of Majocchi after Anti-SARS-CoV-2 Pfizer-BioNTech Vaccine: Is There an Association?"

_vaccines, 2022, doi:10.3390/vaccines10111972_

Round 1

Reviewer 1 Report (Previous Reviewer 3)

The author provided the justification and comments for most of the concerns raised in the previous versions.

Author Response

Dear Reviewer n'1, thank you very much.

A warm greeting.

Reviewer 2 Report (New Reviewer)

Dear Authors

it is an interesting report and it is very honest from you to underline that the a cause-effect correlation between PAT and vaccination has not been proven.

Moreover, you can add which are the risk factors for PAT? Obesity, overweight? Smoke habit? Others? Your patient had other risk factors a part the vaccination?

line 147 " some reports have been reported" should be rephrased

Author Response

Dear Reviewer n'2,

thank you very much.

We added some informations about the etipathogenesis of Majocchi Diseases (PAT) and we added that our patients didn't have any risk factor. 

Reviewer 3 Report (New Reviewer)

Congratulations on the interesting and well documented case report you submitted to the Journal. 

The topic of your paper is not original, but it is useful for a greater understanding of post-vaccination phenomena. 

The main question is well addressed by this study. 

Your article is relevant and interesting. 

Your paper is well written and the text is clear and easy to read. 

The conclusions are consistent with the evidence and arguments presented and they address the main question. 

I appreciate the restraint that shines through in the conclusions.

Author Response

Thank you very much.

This manuscript is a resubmission of an earlier submission. The following is a list of the peer review reports and author responses from that submission.

Round 1

Reviewer 1 Report

The authors reported a clinical case of purpura annularis telangiectodes of Majocchi in Italy. The patient developed disease-relevant symptoms after the Pfizer COVID19 mRNA vaccination. Personally, I think the manuscript suffered from a few major drawbacks. 

(1) It is unclear to readers whether there is a correlation between the vaccination and cutaneous signs. This case study represents a rare event that might or might not be scientifically significant. 

(2) The authors only provided two descriptive pictures and failed to provide any detailed immological characterization data, like complete blood cell count, basic metabolic panel, complement, C-reactive protein, histopathologic examination and etc. 

Author Response

1) The purpose of our study is to encourage the report of other similar cases in order to explore the possible correlation between these two events. In fact, we cannot prove in any way a cause-effect correlation between these two events but, as suggested by other similar cases already published in literature, the temporal correlation, the rapid clinical resolution and the unknown exact pathogenesis of Majocchi disease may increase the suspect for a connection between the vaccination and this dermatological condition. An immune cross-reactivity and a hypersensitivity mechanism could indeed justify the skin manifestations in our patient. Moreover, as the clinical manifestations regressed quite easily in our patient without any recurrences in the follow up, it is possible to think that a temporally circumscribed trigger event, such as a vaccination, induced it. We also want to reinforce the fact that this reaction does not necessarily have to be present after the first dose and with topical therapy can go into remission.

2) In this new version we report new laboratory and histopathological data not included in the previous version and we apologize for the oversight.

Reviewer 2 Report

Case reports are very important when when assessing new developments associated with an emerging treatment such as the SARS-CoV2 vaccine.  Your admonition to not let this type of case report discourage the use of vaccine is much appreciated.

Nonetheless, it would be very interesting to determine what types of lymphocytes are present in the lesions described with special interest in the type of T-lymphocyte that participates in the development of the described skin lesions.

Although there is much more research that needs to be done case reports like this are extremely important to alert health care practitioners.

Author Response

We deeply thank the reviewer, as we are pleased to receive these favorable comments regarding our case report. In the revised version of this manuscript, we added a histological description of our patient’s lesions, with a particular focus on the distribution of the inflammatory infiltrate. We also performed immunohistochemistry in order to better analyze the presence of CD4 and CD8 around the lesions, but as no clear prevalence of one population over another was detected, we did not further investigate this topic.

Reviewer 3 Report

The case report described by the Ambrogio group is interesting and align with some of the allergic reaction reported with the Covid-19 vaccination. Varying from other case reports, the author correlated Majocchi’s Disease with Covid-19 vaccination side effects, especially the Pizer-BioNTech mRNA vaccine. Gaps that exist in this report require further explanation before reaching any conclusion.

I have the following comments about this report:

  • In the past adverse effect of booster doses seen in elder people aged 60 or high. However, in this report patient is 46 years old with symptoms identical to Gougerot Blum Disease, a type of Majocchi's Disease.
  • In this case report, Majocchi's Disease symptoms were seen after 30 days of the third booster dose. Whereas in other cases, adverse effects are seen much earlier.
  • As the actual cause of Majocchi's Disease is not well defined. So immunological profiling is required to establish the correlation with the mRNA vaccine. Along with this, no histological data was provided to support the claim made in this case report.
  • Erythematous annular patches shown in figure 1 are very different from the actual patches observed for Majocchin's Disease. Please refer to figure 1 for articles 1, 2, and 14.
  • The author cited in line 109 that "To the best of our knowledge, only three clinical cases of Majocchi’s disease have been reported as a chronological consequence of vaccination: one after flu vaccination and two after anti-SARS-Cov2 one" with reference 14. In this reference, I could only see one case where a 74 years old man had a severe allergic response to the flu vaccine and examination showed numerous annular and polycyclic macules with purpuric petechiae. The patient also had several other complications with long medical history.

Author Response

We thank the reviewer for the comments. In our article, we do not want to describe Majocchi disease as a negative adverse effect of Sars-CoV-2 vaccination, as we deeply believe that vaccines are essential to prevent severe symptoms and even deaths. This article aims indeed to emphasize the possibility of a link between these two events in order to stimulate scientific discussion and research regarding a possible immune-mediated pathogenic mechanism of Majocchi disease. Moreover, as this dermatological condition generally has a favorable course as demonstrated by our case, we do not discourage Sars-CoV-2 vaccination neither in the general population, nor in patients who may have already exhibited this condition.

As for the timing of the manifestations and as for our patient’s characteristics, we would like to highlight that in 2022 Atak et al described a similar case of PPD in a young male, suggesting that different ages may be at risk of developing this condition. Moreover, although we agree with the reviewer regarding the early onset of adverse effects to vaccines, we would also like to underline a certain variability among patients, as no clear rules have been described yet, especially considering the recent development of these vaccines.    

Moreover, in the revised version of this manuscript, we provided three new histological images and performed immunohistochemistry investigations to better define the lymphocytic infiltrate surrounding the histological sections. However, as no clear prevalence of one population over another has been detected, we decided not to further investigate this aspect.

As for the clinical aspect of our patient’s lesions, figure 3 by Spigariolo et al.3 and both figure 1 and 2 by Atak et al.7, show lesions very similar to the one described in our case. Moreover, figure 2 by Atak et al.7 demonstrate very similar dermatoscopic features to the one shown in our article. 

As for the differential diagnosis with Gourgerot Blum disease, we would like to emphasize that lesions were all flat without any palpable elements, as well as that no lichenoid papules were detected.

We also inserted the two bibliographical notes that had been accidentally removed, describing pigmented purpuric dermatoses like Majocchi disease

Round 2

Reviewer 3 Report

The author tried to provide a modified version with more histological images on different magnifications however without the control images it would be difficult to interpret data. In line 78 author mentioned, "The immunohistochemistry shows CD4 and CD8 lymphocytes in the same proportion". In this image, the author did not specify these lymphocytes and not specified what method they used to differentiate between CD4 and CD8 cells. With this, there are still comments from the previous review that require a scientific explanation.